# Motion along the mental number line reveals shared representations for numerosity and space

Caspar M Schwiedrzik[1]*, Benjamin Bernstein[2], Lucia Melloni[3,4,5]*

[1]Laboratory of Neural Systems, The Rockefeller University, New York, United States; [2]Department of Psychology, Northwestern University, Evanston, United States; [3]Department of Neurophysiology, Max Planck Institute for Brain Research, Frankfurt am Main, Germany; [4]Department of Neurosurgery, Columbia University College of Physicians and Surgeons, New York, United States; [5]Department of Neurology, New York University Langone Medical Center, New York, United States

**Abstract** Perception of number and space are tightly intertwined. It has been proposed that this is due to 'cortical recycling', where numerosity processing takes over circuits originally processing space. Do such 'recycled' circuits retain their original functionality? Here, we investigate interactions between numerosity and motion direction, two functions that both localize to parietal cortex. We describe a new phenomenon in which visual motion direction adapts nonsymbolic numerosity perception, giving rise to a repulsive aftereffect: motion to the left adapts small numbers, leading to overestimation of numerosity, while motion to the right adapts large numbers, resulting in underestimation. The reference frame of this effect is spatiotopic. Together with the tuning properties of the effect this suggests that motion direction-numerosity cross-adaptation may occur in a homolog of area LIP. 'Cortical recycling' thus expands but does not obliterate the functions originally performed by the recycled circuit, allowing for shared computations across domains.

*For correspondence: cschwiedrz@rockefeller.edu (CMS); lucia.melloni@brain.mpg.de (LM)

**Competing interests:** The authors declare that no competing interests exist.

## Introduction

Our perception of numerosity and space are tightly interrelated, as evidenced by the 'mental number line', where small numbers are mapped to the left and large numbers are mapped to the right (*Dehaene et al., 1993*). This spatial arrangement of numbers is evident in preverbal infants (*de Hevia and Spelke, 2010*), non-human primates (*Drucker and Brannon, 2014*), and even birds (*Rugani et al., 2015*), suggesting a deep evolutionary heritage of the mental number line, although there is evidence that it can be influenced by cultural practice (*Shaki and Fischer, 2008*). Number-space interactions are attributed to the fact that the neural bases of numerical and space processing both localize to parietal cortex (*Hubbard et al., 2005*), as evidenced by the fact that parietal lesions disrupt both numerical and spatial processing (*Zorzi et al., 2002*). It has been suggested that this co-localization is the result of 'cortical recycling' during evolution or normal development (*Anderson, 2010*; *Dehaene and Cohen, 2007*), i.e., numerical cognition builds upon circuits that originally process space, because the computations that are used to gauge target coordinates or to transform reference frames can also be used for operations on numbers. But to what extent do such recycled circuits retain their original functionality? Does numerosity replace space, or does it augment it, allowing for shared computations across domains? We describe a new phenomenon in which visual motion direction, a spatial dimension ubiquitous in dorsal stream areas, adapts numerosity, giving rise to a repulsive aftereffect: motion to the left adapts small numbers, leading to overestimation of

**eLife digest** Our sense of number is thought to have emerged from the circuits of cortical neurons in the brain that originally represent space, a process known as 'cortical recycling'. Accordingly, our perception of space and number are tightly intertwined: for example, people think about numbers on a mental number line, where smaller numbers are mapped to the left, and larger numbers are mapped to the right. Also, damage to a brain region called the parietal cortex disrupts both space and number processing.

If number processing recycles the neurons that encode space, which form does this appropriation take? Recycling could preserve the original behavior of the neurons (processing space), thus enriching the neurons' functional repertoire with a new capacity (processing number). Alternatively, the newly developed role could replace the original one, such that space and number cohabitate the same brain area but use separate neurons.

To disentangle these hypotheses, Schwiedrzik et al. used a technique called 'perceptual adaptation'. Here, continuously showing someone a particular feature eventually exhausts the neurons that respond to that feature. Neurons that respond to the opposite feature are however less exhausted and dominate perception. Consequently, people perceive the opposite of what they are adapted to. For example, after continuously seeing dots moving to the right, people perceive stationary dots as moving to the left. Similarly, after being exposed to large numbers of dots they underestimate how many dots they see.

If the same neurons process numbers and space, then adapting to movement in a particular direction should influence number perception. During Schwiedrzik et al.'s experiments, volunteers watched moving dots on a computer screen. After seeing dots move to the right, they underestimated the number of dots that then appeared on the screen. This is likely to be because larger numbers are mentally mapped to the right, and seeing rightward motion for a long time exhausted these neurons. This means that neurons that respond to smaller numbers (mentally mapped to the left) were more active when the new dots were presented, leading the volunteers to underestimate how many dots they saw. Adapting to leftward motion led to the opposite effect, with volunteers overestimating the number of dots. Thus, motion can literally move us up and down the number line.

These results indicate that the same neurons encode both space and numbers. Cortical recycling does not erase the neurons' original behavior: instead, neurons may carry out the same computations when processing numbers or space. This would allow the brain to add new functionality without sacrificing any of the computational resources for processing space.

numerosity, while motion to the right adapts large numbers, resulting in underestimation. Thus, motion can literally move us up and down the mental number line. This indicates that numerosity coopts circuits encoding space. Furthermore, we show that the reference frame of this cross-adaptation effect is spatiotopic, not retinotopic, which, together with the tuning profile of the adaptation effect, indicates that motion direction-numerosity cross-adaptation may occur in a homolog of area LIP. Thus, recycled numerosity processing circuits indeed remain sensitive to purely spatial features. This shows that 'cortical recycling' augments but does not obliterate the spatial capacities of parietal circuits.

## Results

We used an adaptation paradigm – the psychophysicist's microelectrode (*Mollon, 1974*) – to test whether recycled numerical representations retain sensitivity to a spatial feature that is itself orthogonal to magnitude, namely motion direction. Because the parietal circuits that process number are known to also contain neurons that respond to motion (*Colby et al., 1993*; *Fanini and Assad, 2009*), we hypothesized that motion to the left moves us down the number line, while motion to the right moves us up the number line. Classically, adaptation is thought to systematically reduce the responsivity of neurons that encode the adapted feature, leading to an overshoot of activity in neurons that respond to the opposite feature. This typically results in a perceptual aftereffect in which

subjects perceive the opposite of what they have been adapted to (*Solomon and Kohn, 2014*). For example, in the classical motion aftereffect (MAE), subjects perceive leftwards motion after being adapted to rightwards motion (*Levinson and Sekuler, 1976*), and in numerosity adaptation, subjects underestimate the numerosity of dots after being adapted to large quantities (*Burr and Ross, 2008*). While those studies have revealed adaptation *within* a domain, either motion or number, here we test for adaptation *across* domains. In particular, we hypothesized that if the same population of neurons coded for motion direction and number, adaptation to leftwards motion should lead to subsequent overestimation of numerosity, while adaptation to rightwards motion should lead to underestimation, reflecting a repulsive aftereffect. In contrast, if number and motion direction were coded by different neurons, a cross-adaptation effect would not be expected, even if those neurons co-localized to the same brain area.

In a first experiment, we adapted 11 subjects to two concurrently presented displays of 400 dots. The dots in the upper display were moving either leftward or rightward, while the dots in lower display were moving randomly. We then asked subjects to indicate with a button press which of two subsequently presented clouds of randomly moving dots, the 'test' or the 'probe', was more numerous (*Figure 1*). We obtained psychometric functions by varying the number of dots (23–107) in the test cloud, which was shown at the location adapted to motion direction, while holding the number of dots in the probe cloud, which was only adapted to numerosity, but not motion direction, constant (30). This allowed us to determine the point of subjective equality (PSE), i.e., the number of dots at which the test and probe appear to have the same numerosity, for each motion direction, and thus to quantify whether motion direction affects numerosity perception. Because the PSE arises from the comparison between two locations that exclusively differ in whether they are adapted to motion direction or not, effects of static numerosity (*Burr and Ross, 2008*) and/or texture density (*Durgin, 2008*) cannot explain any differential effect of motion direction.

*Figure 2a* shows that adaptation to motion direction indeed resulted in a cross-adaptation effect on numerosity: Adaptation to rightward motion led subjects to perceive the test cloud as relatively less numerous than adaptation to leftwards motion (mean difference in PSE 7.12 dots, $t(10)$=2.555, p=0.029, $d_{Hedges}$=0.953), in accordance with a repulsive aftereffect. This effect, which was clearly evident in individual subjects (*Figure 2a,b*; *Figure 2—figure supplement 3*), shows that exposure to motion direction leads to over- and underestimation of nonsymbolic quantities, as if motion moves us up and down the number line. Motion direction exclusively affected the PSE but not psychometric slopes (mean difference -0.009, $t(10)$=−1.038, p=0.324, $d_{Hedges}$=−0.307). Note that all psychometric functions are shifted to the right because of strong, inevitable adaptation effects of large numerosities (*Burr and Ross, 2008*) and/or texture density (*Durgin, 2008*) that cause subjects to underestimate consecutively presented dot clouds, while the distance between leftward and rightward motion reflects the differential effect of motion direction on numerosity perception.

Cross-adaptation between motion direction and numerosity also occurred when we adapted with small numbers of dots: When adapting with 30 dots and probing with 166 dots (instead of adapting with 400 dots and probing with 30 dots) in Experiment 2, we found that, as in Experiment 1, the PSE after leftward motion adaptation was smaller than after rightward motion adaptation (*Figure 2c*; mean difference in PSE 15.85 dots; $t(9)$=4.523, p=0.001, $d_{Hedges}$=1.017). Again, there were no effects on slopes (mean difference −0.123, $t(9)$=−0.755, p=0.469, $d_{Hedges}$=−0.313). Hence, motion direction has the same differential effects on numerosity perception when adapting with small (30 dots) and when adapting with large quantities (400 dots). In addition, all psychometric functions were now shifted to the left of the probe, reflecting that adaptation with small numerosities generally leads subjects to overestimate consecutively presented dot clouds (*Burr and Ross, 2008*). Together, this indicates that leftward motion more strongly adapts neurons coding for small numbers than rightward motion, causing a stronger repulsive aftereffect and thus increased overestimation.

In Experiments 1 and 2, we used an adaptation paradigm to reveal the effects of motion direction on numerosity perception. However, if the same neurons indeed encode both numerosity and motion direction, then motion direction should affect numerosity perception also in the absence of adaptation. Experiment 3 (*Figure 2d*) showed that the interaction between motion direction and numerosity perception was also evident when subjects *directly* judged the numerosity of coherently moving dots, without a preceding adaptation phase: Rightward motion led to an *overestimation* of numerosity (a leftward shift of the PSE), while leftward motion led to *underestimation* (a rightward

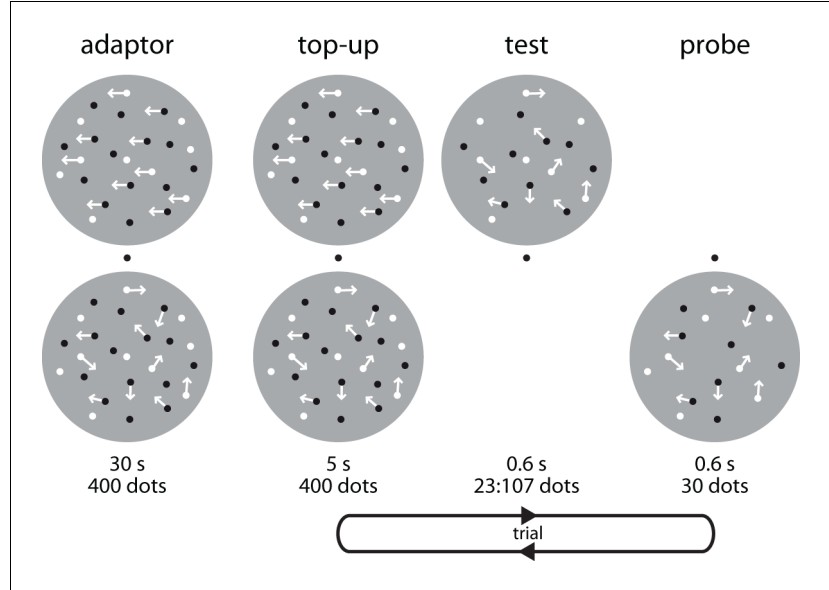

**Figure 1.** Cross-adaptation paradigm to test the effect of motion direction on numerosity perception. Each *block* started with a 30 s adaptor, consisting of two clouds of 400 dots each (50% black, 50% white). Dots in the bottom cloud always moved randomly (0% coherence). Dots in the top cloud moved leftward or rightward at 100% coherence (as indicated here by arrows). To maximize directional adaptation at the top location, each condition was tested in a separate session. Each *trial* started with a 5 s top-up adaptor. Dots in the bottom cloud would again always move in random directions. Dots in the top cloud would move rightward or leftward at 100% coherence. After a 400 ms inter-stimulus interval (ISI), we presented a 'test' stimulus consisting of randomly moving dots at the top location. After another 400 ms ISI, we presented the 'probe' stimulus, also consisting of randomly moving dots, at the bottom location. In Experiment 1, the 'test' contained 23:107 and the probe contained 30 randomly moving dots. Subjects indicated which cloud, the 'test' or the 'probe', was more numerous by pressing a button on a keyboard. This paradigm allowed use to investigate motion direction-numerosity cross-adaptation and also replicated the well-known numerical distance effect (*Figure 1—figure supplement 1*). We used the same basic task structure also in Experiments 2 and 6, but changed the number of dots in the adaptors, test, and probe stimuli.

The following figure supplement is available for figure 1:

**Figure supplement 1.** Numerical distance effect.

shift of the PSE) (mean difference right vs. left PSE -2.08, $t(9)=-2.47$, $p=0.035$, $d_{Hedges}=-0.707$). Since there was no adaptation, this *direct* effect is of the opposite sign than the repulsive aftereffect reported in Experiments 1 and 2. The smaller effect size than in the adaptation experiments is likely due to the shorter exposure to motion direction (600 ms vs. 5000 ms). There were no effects on slopes (mean difference 0.007, $t(9)=1.001$, $p=0.342$, $d_{Hedges}=0.351$).

Together, Experiments 1–3 demonstrate that motion direction affects the perception of large as well as small numerosities, moving us up and down the number line, and that this interaction can be revealed by repulsive aftereffects but does not depend on them, as it is also observed directly, i.e., in the absence of adaptation.

For the adaptation studies, there are however two alternative scenarios by which motion direction could affect numerosity: motion direction could either directly act on numerosity, as predicted by an account in which the same circuits process number and motion, or the effect could arise from the perception of illusory motion in the test clouds as a consequence of a classical MAE. Such illusory motion could, for example, draw attention in the opposite direction of the adapting motion stimulus, thus shifting subjects' focus of attention up or down the number line. To rule out this alternative interpretation, we directly assessed the presence of classical MAEs in our paradigm in Experiment 4. None of the 11 subjects we tested perceived illusory motion in the test/probe stimuli (*Figure 2—figure supplement 1* and *Figure 2—figure supplement 2*). This demonstrates that the cross-

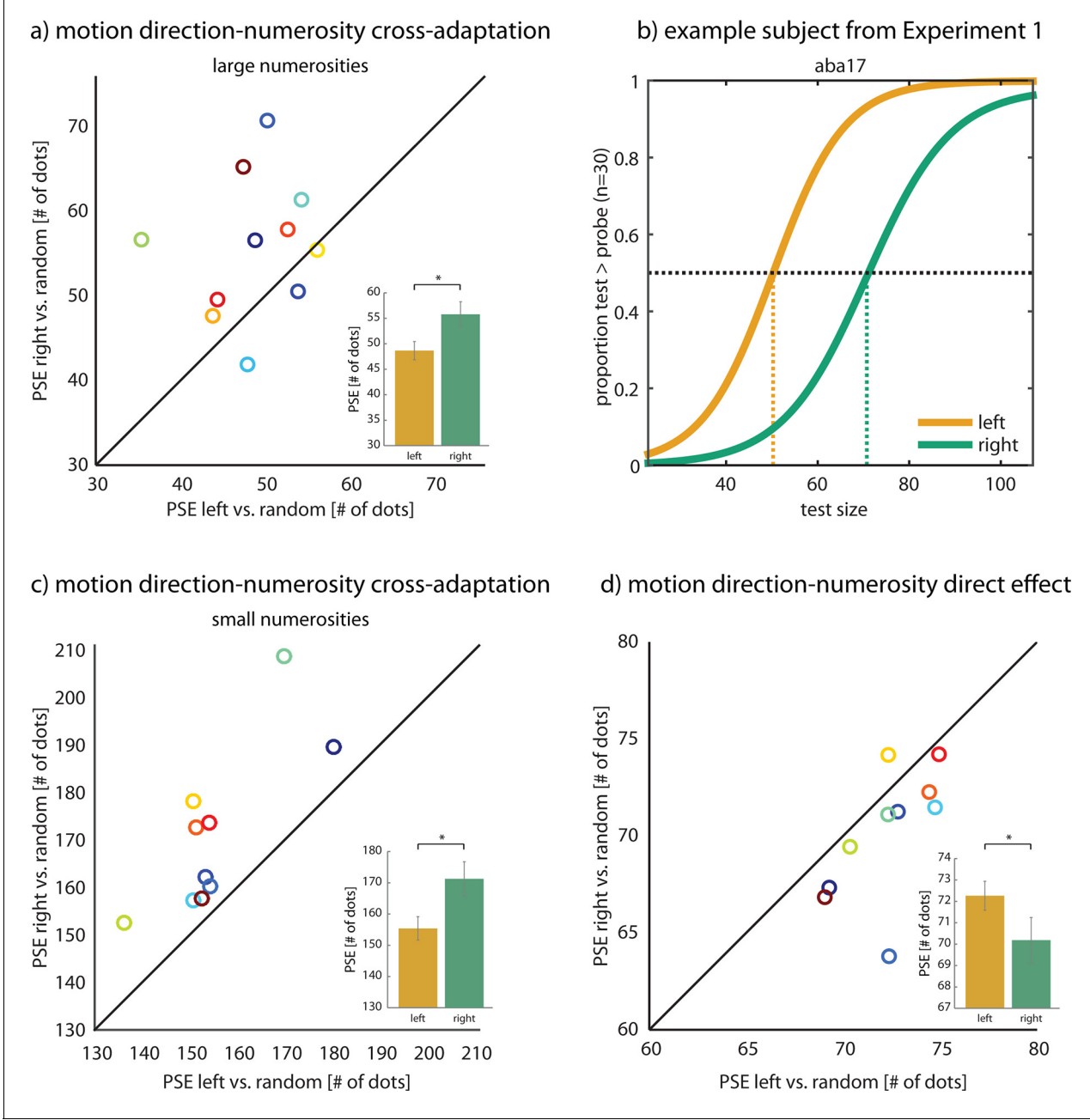

**Figure 2.** Effect of motion direction on numerosity perception. (a) There was a significant repulsive aftereffect of motion direction (left, right) on perceived numerosity (mean difference in PSE 7.12 dots, *t*(10)=2.555, p=0.029, *d*Hedges=0.953; see inset) when adapting with large numerosities (400 dots) in Experiment 1. The scatter plot pits the PSE after rightward motion against the PSE after leftward motion per subject. It shows that adaptation to rightward motion led subjects to perceive the test cloud as less numerous than adaptation to leftward motion, as evidenced by the large fraction of dots above the equality line, which is indicative of a repulsive aftereffect. This was not due to a classical motion aftereffect (*Figure 2—figure supplement 1* and *Figure 2—figure supplement 2*). (b) Psychometric functions of subject aba17. After adaptation to 400 rightward moving dots in Experiment 1, this subject perceived 70 dots in the test dot cloud to be equivalent to 30 dots in the probe dot cloud, thus underestimating the number of dots in the test. After adaptation to leftwards motion, aba17's PSE was 20 dots lower than after adaptation to rightwards motion, although the number of dots in the two clouds were identical in both conditions. Note that all psychometric functions are shifted away from 30 dots, the probe size, because of a static numerosity adaptation effect that causes subjects to underestimate consecutively presented dot clouds (*Burr and Ross, 2008*). See *Figure 2—figure supplement 3* for additional example subjects. (c) There was also a repulsive motion direction-numerosity cross-adaptation effect after adaptation to small numerosities (30 dots) in Experiment 2. When adapting with numerosities smaller than the probe (166 dots), subjects overestimate the number of dots in the test cloud (*Burr and Ross, 2008*). Leftward motion-numerosity cross-adaptation should exaggerate this overestimation effect relative to rightward motion. Indeed, the scatter plot shows that PSEs after leftward motion were consistently smaller than for
*Figure 2 continued on next page*

*Figure 2 continued*

rightward motion (mean difference 15.85 dots; $t(9)$=4.523, p=0.001, $d_{Hedges}$=1.017, two-sided). This indicates that, as when adapting with large numerosities, leftward motion shifts numerosity perception down the number line, while rightwards motion shifts numerosity perception up the number line. (**d**) Motion direction affected numerosity perception also directly, in the absence of adaptation. In Experiment 3, subjects compared the numerosities of coherently and incoherently moving dot clouds without prior adaptation. The scatter plot pits the PSE for rightwards motion against the PSE for leftwards motion per subject, and shows that, indeed, all but one subject perceived clouds of rightward moving dots as more numerous than randomly moving clouds (0% coherence) and clouds of leftward moving dots as less numerous than randomly moving clouds (0% coherence). Accordingly, the PSEs for rightwards vs. leftwards conditions were significantly different (mean difference right vs. left −2.08, $t(9)$=−2.47, p=0.035, $d_{Hedges}$=−0.707, two-sided). Thus, motion direction affects numerosity estimates also without a preceding adaptation phase. Note that since there is no adaptation, there is also no repulsive aftereffect and hence, the direct effect is of the opposite sign than the repulsive aftereffects in a and c. Data in insets are represented as mean ± SEM. All data shown here are publicly available at Figshare (*Schwiedrzik et al., 2015*).

The following figure supplements are available for figure 2:

**Figure supplement 1.** Adaptation paradigm to test classical motion aftereffects.

**Figure supplement 2.** Control for classical motion aftereffects.

**Figure supplement 3.** Additional example subjects from Experiment 1.

adaptation aftereffect on numerosity directly arises from motion direction adaptation, and is - like other high-level MAEs (*Whitney and Cavanagh, 2003*) - not mediated by a classical MAE. This also speaks against the notion that attention could be drawn in the direction of a repulsive classical MAE during the presentation of the test cloud, thus shifting the focus of attention up or down the number line.

Together, Experiments 1–4 indicate that recycled numerosity circuits continue to process purely spatial features. Where could this effect arise? In monkeys, the first stage of number processing is in the intraparietal sulcus (IPS), specifically in areas LIP and VIP (*Nieder and Dehaene, 2009*). Comparing our psychophysical results to the known numerosity tuning functions of these two areas gives a first indication of where and how this cross-adaptation effect may come about (*Figure 3*): Because we were able to exert an adaption effect despite a large difference between the number of adapting dots (e.g., 400 in Experiment 1) and the number of dots in the test/probe display (e.g., 23:107; 30 in Experiment 1), the tuning functions of the involved neurons must be rather broad. Such broad tuning curves for numerosity have been found in monkey area LIP, where neurons show monotonic increases and decreases in firing rate with the number of items in a display (*Roitman et al., 2007*) and are also sensitive to motion direction (*Fanini and Assad, 2009*). LIP is thought to correspond to an intermediate analog representation of numerosity before the cardinal representation of number arises (*Dehaene and Changeux, 1993*; *Verguts and Fias, 2004*) downstream in area VIP, which is also sensitive to motion (*Colby et al., 1993*) but where numerosity neurons show much narrower tuning functions (*Nieder and Dehaene, 2009*). If adaptation arose in VIP, the strongest effects should occur directly around the adaptor, but fall off the further the adaptor is removed from the test/probe (*Robbins et al., 2007*; *Webster, 2011*). In contrast, we found that adapting and probing with the same number of dots (30 in Experiment 5) did not lead to a significant motion direction-specific aftereffect (mean difference right vs. left −3.69, p=0.92, Wilcoxon signed rank test, one-sided; Bayesian posterior probability <0.18%). The presence of a strong adaptation effect when adapting with 400 and testing with 30 dots and the absence of an adaptation effect when adapting and testing with 30 dots together indicate that motion direction-numerosity cross-adaptation may arise in a human homolog of area LIP.

## The reference frame of motion direction-numerosity cross-adaptation

A second characteristic of the neural circuitry in the IPS is that it serves to convert retinotopic coordinates prevalent in early visual areas into eye-, head-, and world-centered frames of reference. In particular, areas LIP and VIP both contain neurons with eye- and head-centered coordinates (*Duhamel et al., 1997*; *Mullette-Gillman et al., 2005*). However, the two areas can be distinguished based on their respective receptive field properties: while neurons in area LIP have spatially

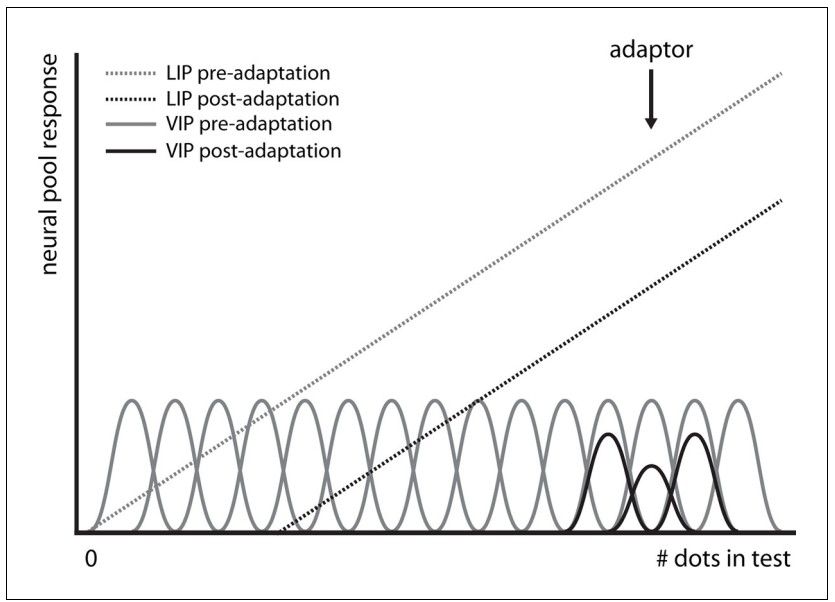

**Figure 3.** Hypothetical effect of adaptation on numerosity-encoding neurons in LIP and VIP. It has been shown that area LIP (dotted lines) contains neurons that encode numerosity with monotonically increasing firing rates, i.e., the larger the number of dots in the display, the more vigorous the neural response (*Roitman et al., 2007*). In contrast, one stage downstream from area LIP, neurons are narrowly tuned to numerosity in area VIP (solid lines), i. e., each neuron will respond vigorously only to a narrow range of numerosities, and less so when either more or less dots are displayed (*Nieder et al., 2006*). Adaptation has different effects on these different classes of tuning curves (black lines): In area VIP, adaptation will only affect neurons that encode the number of dots in the adapter (400 in Experiment 1) while neurons coding for numerosities far removed from the adapter, such as the ones in the test (23:107 in Experiment 1) and probe dot clouds (30 in Experiment 1), will not be affected. In contrast, in area LIP, adaptation will shift the entire tuning curve, hence even adaptation to 400 dots affects responses to numerosities in the range of 23:107. Thus, the adaptation effect we observe is likely to arise in area LIP, not VIP.

restricted receptive fields, each covering only a portion of the visual world, number-sensitive neurons in area VIP, the next stage of number processing, respond to stimuli distributed over the entire visual field (*Nieder and Dehaene, 2009*). We capitalized on these distinct properties of LIP and VIP and tested in Experiment 6 whether the cross-adaptation effect between motion direction and number occurs in a retinotopic (early visual areas) or spatiotopic (IPS) frame of reference, and whether it is spatially specific (LIP) or occurs over the entire visual field (VIP), thus allowing us to infer the cortical area in which motion and number processing interact.

To this end, in Experiment 6 we replicated Experiment 1 but now also manipulated the location of the test/probe stimuli on the screen relative to subjects' gaze (*Figure 4*): Subjects made two saccades between the adaptor and the test/probe stimuli. The second saccade would either bring their gaze back to the initial fixation location (*Figure 4a and d*), or to a new location 14 degrees visual angle (dva) away from the original center of gaze (*Figure 4b and c*). We then presented the test/probe stimuli either at the same location on the screen as the adaptor (*Figure 4b*), at the same location as the adapter relative to the subject's current gaze (*Figure 4c*), or at a new location (*Figure 4d*). This allowed us to compare two frames of reference and the spatial specificity of the cross-adaptation effect. If the cross-adaptation effect was present only when subjects looked at a new location on the screen but the stimuli were presented at the same position relative to their current center of gaze (*Figure 4c*), it must take place in areas with a retinotopic frame of reference, i.e., early visual areas; if cross-adaptation was evident only when subjects fixated at a new location but the stimuli were presented at the original location on the screen (*Figure 4b*), it must occur in an area with a spatiotopic frame of reference and with restricted receptive fields, such as LIP; and if it occurred when subjects looked at the original fixation spot but stimuli were presented at a location that neither matched retino- nor spatiotopic reference frames (*Figure 4d*), the area in which cross-

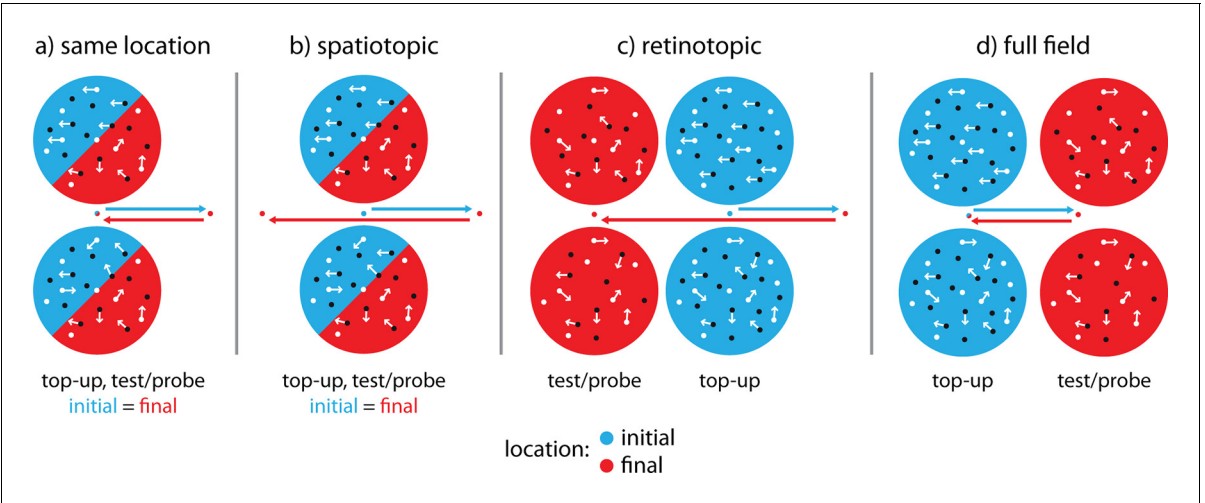

**Figure 4.** Paradigm to test cross-adaptation in different reference frames. The design of Experiment 6 closely followed that of Experiment 1, but introduced two saccades (arrows) between the top-up adaptor and the test/probe stimuli. Initial stimulus and fixation locations are depicted in blue, final stimulus and fixation locations in red. (a) In the 'same location' condition, subjects made a 14 dva leftward or rightward saccade after the top-up adaptor had been presented, and then back to the original fixation dot. The location of the test/probe stimuli and adaptor was the same in space and on the retina, depicted here as blue (initial) and red (final) half-cycles. This condition served to replicate the cross-adaptation effect found in Experiment 1. (b) In the 'spatiotopic' condition, subjects made two consecutive leftward or rightward saccades (first 14 dva, second 28 dva), and test/probe stimuli were presented at the same screen position as the top-up adaptor but on a different retinal position. Here, a cross-adaptation effect would occur in an area that has spatiotopic but locally restricted receptive fields, such as LIP. (c) In the 'retinotopic' condition, subjects also made two consecutive leftward or rightward saccades (first 14 dva, second 28 dva) but test/probe stimuli were presented above and below the new fixation dot. Hence, stimuli occurred at the same retinal location (relative to the subjects' gaze), but at different location in spatiotopic coordinates. Here, the cross-adaptation effect would occur in early, retinotopically organized visual areas. (d) In the 'full field' condition subjects first made rightward or leftward saccaded (14 dva), and then back to the original fixation dot (14 dva, as in the 'same location' condition). Test/probe stimuli were however presented at a new location opposite to the second saccade direction. Here, a cross-adaptation effect would only occur if the effect takes place in an area with very large receptive fields, such as VIP.

adaptation takes place must support effects across the entire visual field, as would be predicted for VIP. *Figure 5* shows that cross-adaptation between motion direction and number occurs even in the presence of two intervening saccades, replicating Experiment 1. More importantly, the results of Experiment 6 show that cross-adaptation takes place in a spatiotopic, not retinotopic frame of reference, but is restricted to the spatiotopic location of the stimulus and does not extend over the full field of view (motion direction × reference frame interaction, $F(3,15)=4.58$, $p=0.018$, $\eta^2=0.478$). Only the 'same location' and the 'spatiotopic' condition were significantly different from zero (both $p=0.015$, Wilcoxon signed rank tests), while the 'retinotopic' and 'full field' condition were not ($p=0.71$ and $p=0.84$, respectively). Bayesian statistics (*Benavoli et al., 2014*) confirmed that there was overwhelming evidence for an effect in the 'same location' and 'spatiotopic' (posterior probabilities >95%), but not in the retinotopic (posterior probability <44.5%) or full field (posterior probability <29.8%) conditions. Retinal eccentricity cannot explain our results, since it was matched in the 'same location' and the 'retinotopic' conditions (0 dva), and in the 'spatiotopic' and 'full field' conditions (14 dva), respectively, and thus orthogonal to the factor reference frame. Instead, spatiotopy allots the effect to a family of MAEs which are thought to tap into higher-order motion processing stages, such as the positional motion aftereffect (*Turi and Burr, 2012*), in contrast to the classical MAE, which occurs in a retinotopic frame of reference (*Knapen et al., 2009*). This pattern of results is predicted by cross-adaptation occurring in a human LIP homolog.

## Discussion

Taken together, our results show on purely behavioral grounds that the neural circuits recycled to process number retain sensitivity to a visual feature intimately linked to our sense of space, namely motion direction. This indicates that 'cortical recycling' expands the original functionality of a circuit

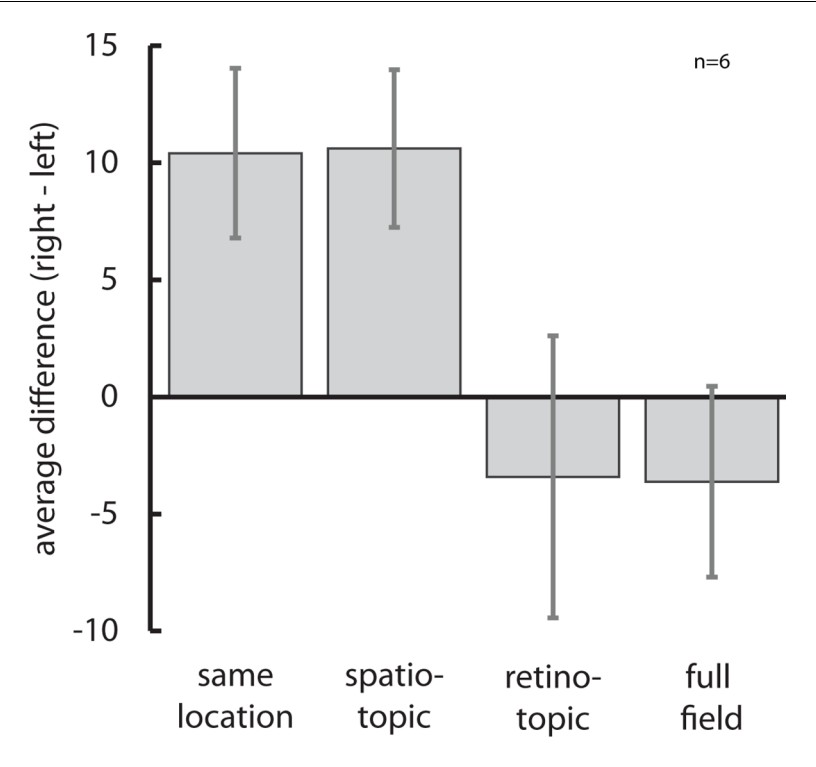

**Figure 5.** The reference frame of motion direction-numerosity cross-adaptation. A non-parametric rmANOVA showed that there was a significant interaction between reference frame and motion direction ($F$(3,15)=4.58, p=0.018, $\eta^2$=0.478). Only the 'same location' condition which replicated Experiment 1 with intervening saccades and the 'spatiotopic' condition with restricted receptive field sizes as predicted for area LIP showed a significant (both p<0.05) cross-adaptation effect from motion-direction on numerosity (see text for statistics). Together with the tuning function of the effect, this indicates that the influence of motion direction on numerosity takes place at an intermediate, analog stage of number processing, and not in early visual areas with a retinotopic frame of reference or in area VIP with receptive fields covering the entire visual field. Data are represented as mean ± SEM. All data shown here are publicly available at Figshare (*Schwiedrzik et al., 2015*).

instead of replacing it, thus establishing a functional link across domains. The combination of properties we find in our study, namely the tuning function, receptive field size, and frame of reference of the motion direction-numerosity cross-adaptation effect matches the known properties of area LIP neurons at the first stage of number processing, which are broadly tuned to numerosity (*Roitman et al., 2007*), have relatively small receptive fields (*Ben Hamed et al., 2001*), and a spatiotopic frame of reference (*Mullette-Gillman et al., 2005*). In contrast, neurons in area VIP, the next stage of number processing, are narrowly tuned and have large receptive fields (*Nieder and Dehaene, 2009*), and neurons in area MT, a preceding stage of motion processing, have a retinotopic frame of reference (*Born and Bradley, 2005*). Hence, a human LIP homolog in the IPS seems to be the most likely origin of the motion direction-numerosity cross-adaptation effect.

Motion direction-numerosity cross-adaptation is consistent with studies showing that transcranial magnetic stimulation (TMS) over parietal cortex can disturb number comparisons and motion detection (*Salillas et al., 2009*), as well as those showing that the number of items in a display can be overestimated when they move at high speed (*Afraz et al., 2004*). However, our studies extend these findings in two critical ways: First, by showing that the very same circuits, and not only the same brain area targeted with TMS, process number and motion; and second, by showing that sensitivity of number processing extends to a spatial feature that is orthogonal to magnitude, namely motion direction (motion speed is a magnitude itself). The latter rules out that motion direction-numerosity cross-adaption is simply the consequence of the operation of a 'generalized magnitude processing system' (*Walsh, 2003*). Furthermore, as our results were not effector-dependent, they strongly suggest that number-space associations occur at the level of the representation, not sensorimotor mapping, which is another function of parietal cortex (*Fogassi and Luppino, 2005*) and has

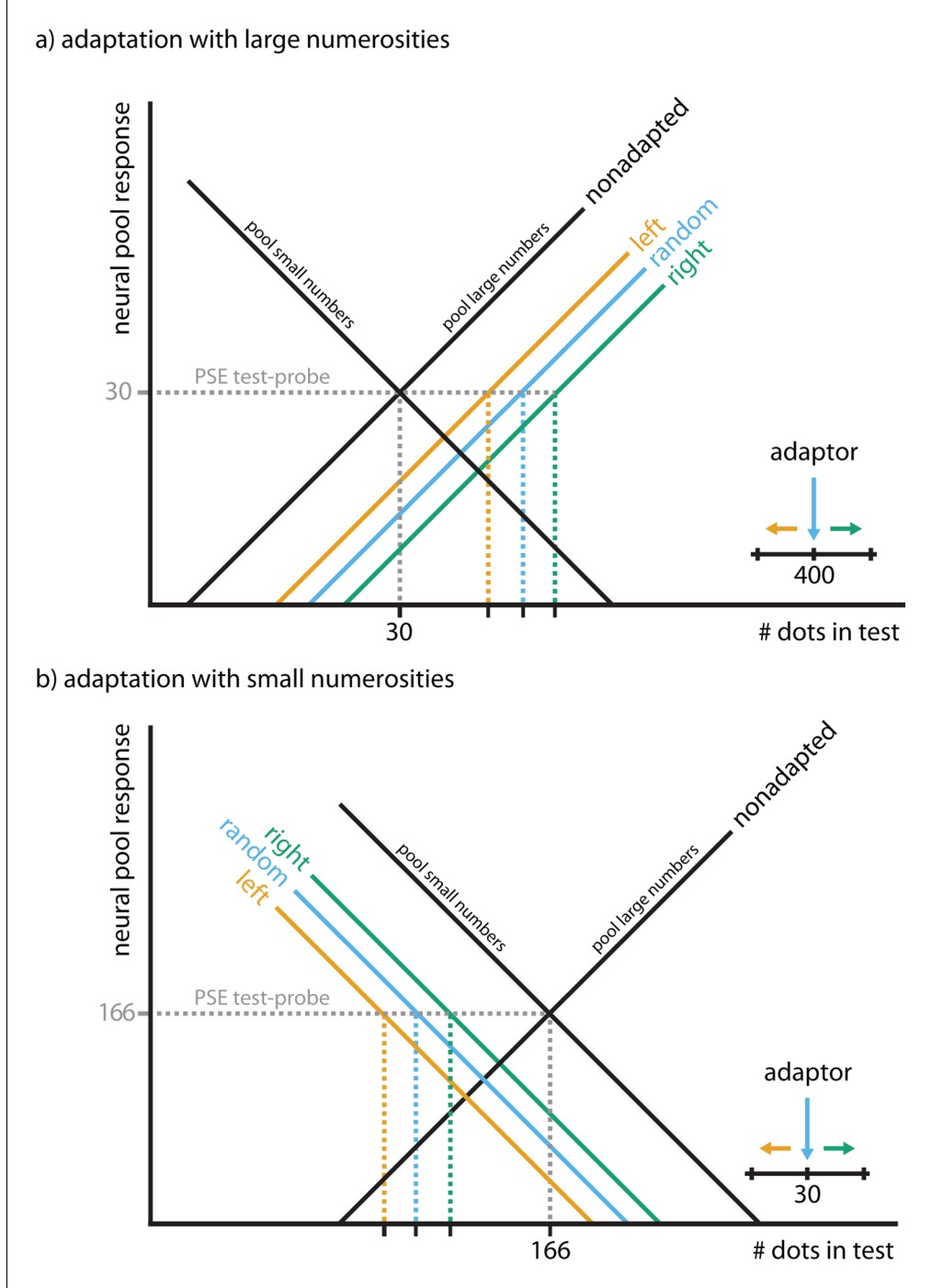

**Figure 6.** Hypothetical model of motion direction-numerosity cross-adaptation. We can explain the cross-adaptation effect using a two-pool opponent coding model similar to what is often invoked to explain the classical MAE and other aftereffects (*Mather and Harris, 1998*). Panel (**a**) exemplifies the effect of motion direction during adaptation with large numerosities (Experiment 1, *Figure 2a*), panel (**b**) exemplifies the effect of motion direction during adaptation with small numerosities (Experiment 2, *Figure 2c*). The two black lines reflect the activity level of two broadly tuned pools of neurons that respond maximally to small or large numbers, respectively. The tuning curves are overlapping such that any given stimulus produces activity in both pools. Adaptation to number causes each pool to reduce its firing rate proportionally to how well the stimulus activates the respective pool in the non-adapted state, e.g., static or randomly moving dot clouds with large numbers of dots (as in Experiment 1, e.g., in the lower cloud, or in *Burr and Ross, 2008*) lead to stronger adaptation in the pool for large numbers (blue). After adaptation, this pool's response is reduced, shifting the point at which the two tuning curves intersect to the either to the right (**a**), i.e., towards larger numbers, or to the left (**b**), i.e., towards smaller numbers. This also shifts the point of subjective equality (PSE) in a comparison task (gray), leading to an underestimation (**a**) or overestimation (**b**) in how many dots are perceived. During adaptation with large numerosities, rightwards motion exaggerates underestimation (**a**) by activating the pool for large numbers

*Figure 6 continued on next page*

*Figure 6 continued*

more strongly, moving perception up the number line (green, inset), subsequently causing a stronger cross-adaptation effect (underestimation). In contrast, leftwards motion reduces the effect, moving perception down the number line (orange, inset), subsequently causing a cross-adaption effect in the opposite direction. During adaptation with small numerosities, leftwards motion exaggerates overestimation (b) by activating the pool for small numbers more strongly, moving perception down the number line (green, inset), subsequently causing a stronger overestimation effect. Rightwards motion reduces overestimation, moving perception up the number line (orange, inset). Monotonically increasing and decreasing tuning curves are consistent with numerosity tuning curves measured in monkey LIP (*Roitman et al., 2007*) and with the fact that we could exert an effect on the perception dot numerosities far removed from the adaptor, e.g., perception of 23–107 dots when adapting with 400 dots in Experiment 1, which would not be expected if each number was represented by narrow tuning curves as those found in monkey VIP (*Nieder et al., 2006*). This was also supported by Experiment 5 which showed that there was no motion direction-numerosity cross-adaption when the number of dots in the adapter and the probe were matched (mean difference right vs. left −3.69, p=0.92, Wilcoxon signed rank test, one-sided; Bayesian posterior probability <0.18%), as predicted for two-pool opponent coding models (*Robbins et al., 2007*; *Webster, 2011*).

been invoked as an alternative explanation for number-space associations (*Keus and Schwarz, 2005*). Motion direction-numerosity cross-adaption can also not be easily explained by attention. Because moving stimuli can attract attention in their direction of motion (*Mattingley et al., 1994*; *Shi et al., 2010*), it is thinkable that attention was drawn down the number line during leftward motion adaptation, and up the number line during rightward motion adaptation. The consequence of this would however be the *opposite* pattern of results from what we found, namely underestimation after leftward motion (because attention is oriented down the number line) and overestimation after rightward motion (because attention is oriented up the number line). Experiment 4 also ruled out that attention was drawn to the left or to the right by illusory motion arising from a classical MAE (*Figure 2—figure supplement 1* and *Figure 2—figure supplement 2*). Hence, attention is an unlikely explanation for motion direction-numerosity cross-adaptation.

Our results are indicative of an opponent coding model such as the one proposed for the classical MAE (*Mather and Harris, 1998*), in which large and small numerosities compete (*Figure 6*). Here, adaptation causes broadly tuned neurons coding for large numbers to reduce their responsiveness, giving rise to an overshoot in the activity of neurons that code for small numbers, and vice versa, leading to under/overestimation. This is evident in the general rightward shift of PSEs reported in Experiment 1 and the leftward shift of PSEs reported in Experiment 2, and consistent with previously reported adaptation effects of static numerosity (*Burr and Ross, 2008*). Our model also captures the co-tuning of leftward motion and small numbers/rightward motion and large numbers: leftward motion increases the activity in the small number pool, resulting in increased overestimation after adaptation, while rightward motion increases the activity in the large number pool, resulting in increased underestimation after adaptation. This causes the differential shifts of PSEs with motion direction along the mental number line.

Opponent coding further suggests that large adaptor – probe/test distances should lead to large repulsive aftereffects as they maximize the differential response ratio between the two pools, while models with multiple, narrowly tuned channels would predict the opposite (*Robbins et al., 2007*; *Webster, 2011*). Specifically, if narrowly tuned neurons such as those found in area VIP (*Nieder et al., 2006*) are adapted, the strongest effects of adaptation occur directly around the adaptor, while effects fall off as a function of adaptor-test/probe distance. This is characteristic of multichannel models. In contrast, in two-pool opponent coding with broadly tuned neurons such as those found in area LIP (*Roitman et al., 2007*), adaptation at the probe location leads to no or only very small effects (assuming the probe stimulus serves as an explicit reference point), as adaptation influences the activity in both pools approximately equally, thus merely reinforcing the current level of overall adaptation. Indeed, we found that large adaptor – probe/test distances led to large repulsive aftereffects, while adapting and probing with the same number of dots did not lead to a significant motion direction-specific aftereffect. This further speaks for two-pool opponent coding and against multichannel models underlying numerosity adaptation. Together, this suggests that current theoretical models of number processing (e.g., *Dehaene and Changeux, 1993*) should be expanded to include such an opponent process at the analog summation stage, which would be compatible with the broad neural tuning curves found in monkey area LIP (*Roitman et al., 2007*).

Finally, it has been shown that the approximate number sense predicts later mathematical achievement (*Halberda et al., 2008*). If the link between motion and number can be further

substantiated, it may open the venue to utilizing motion processing as a marker for number skill development (*Sigmundsson et al., 2010*), and thus to identify children at risk for dyscalculia early in life.

## Materials and methods

### Subjects

Fifty-five subjects participated in six experiments: In *Experiment 1* we investigated motion direction-numerosity cross-adaptation with large numerosities in 11 subjects (4 female, mean age 22.8 years, range 19–33 years, 9 right handed). In *Experiment 2*, we tested for motion direction-numerosity cross-adaptation with small numerosities in 11 subjects (4 female, mean age 27.1 years, range 22–35 years, 9 right handed); one subject had to be excluded from data analyses because of an apparent failure to do the task. *Experiment 3* addressed direct effects of motion direction on numerosity perception in the absence of adaptation in 10 subjects (5 female, mean age 32.2 years, range 18–48 years, 9 right handed). *Experiment 4* served as a control for classical MAEs in 11 subjects (6 female, mean age 33.45 years, range 23–48 years, 10 right handed). *Experiment 5* allowed us to contrast different coding schemes, i.e., two-pool opponent process vs. multichannel models in 6 subjects (3 female, mean age 29.8 years, range 22–39 years, 5 right handed). *Experiment 6* was conducted to probe the reference frame of motion direction numerosity cross-adaptation in 6 subjects (2 female, mean age 23.1 years, range 21–30 years, all right handed). All subjects had normal or corrected-to-normal vision, reported no history of neurological or psychiatric disease, and gave written informed consent before participation. All procedures were approved by The University Committee on Activities Involving Human Subjects at New York University. Subjects received monetary compensation for their participation.

### Stimuli and tasks

Stimulus presentation and response collection in all experiments were controlled using Presentation (v17.2; Neurobehavioral Systems, Berkeley, CA, USA).

### Experiment 1: Motion direction number cross-adaptation with large numerosities

We evaluated whether numerosity perception depends on motion direction using a forced-choice comparison task in which we adapted subjects to leftward and rightward motion, and then tested for motion direction-numerosity cross-adaptation. Each motion direction was presented 200 times over the course of four consecutive blocks, respectively. The order of conditions was pseudo-randomized across subjects. To attenuate carry-over effects between leftward and rightward motion adaptation, subjects completed four blocks during which they were adapted to random motion (0% coherence) between the two directional conditions. Subjects were seated in a darkened experimental booth with their head positioned on a chin rest and fixated at a green fixation dot at the center of the gray screen at 74 cm distance while we presented two clouds (radius 4.1 dva) of moving dots (50% black, 50% white; size 0.04 dva; life time 166.6 ms; constant speed 4 dva/s), positioned symmetrically at 5.5 dva above and below the fixation cross (*Figure 1*). At the beginning of each block, we presented a 30 s adaptor, with 400 dots in the top cloud moving either to the left or to the right at 100% coherence, respectively; 400 dots in the lower cloud always moved in random directions (0% coherence) with the same speed and life time as the dots in the 100% coherence clouds. In the left and right motion direction conditions, the upper location was thus adapted to directional motion, while the lower location was only adapted to the number of dots itself. Hence, by experimental design, PSEs for left and right motion directions subsume the difference between coherent directional and incoherent non-directional motion adaptation (left vs. random, right vs. random). On each trial, we first presented a top-up adaptor for 5 s, in which 400 dots in the top cloud would move in the same direction (left, right) as during the initial adaptation phase, while 400 dots in the bottom cloud again moved randomly. After a 400 ms inter-stimulus interval (ISI), we presented a test stimulus at the top location for 600 ms. The test stimulus contained 23, 27, 32, 38, 45, 54, 64, 76, 90, or 107 randomly moving dots (0% coherence). After another 400 ms ISI, we presented a 600 ms probe stimulus containing 30 randomly moving dots at the bottom location. Subjects had to indicate

with a button press which of the two clouds (top, bottom) was more numerous by pressing the up arrow for the top or the down arrow for the bottom cloud with their right hand on a keyboard. Because subjects are comparing two locations that are both equally adapted to numerosity, the PSE arising from this comparison is relative to the subject's adapted percept of the probe cloud, not the physical number of dots in the probe. We used randomly moving and not static dots to tap into higher-order motion processing stages (*Hiris and Blake, 1992*). Numerosity adaptation is known to be robust against short gaps between test and probe stimuli (*Burr and Ross, 2008*; *Liu et al., 2013*), and importantly, these gaps were identical in all conditions and thus cannot explain differences in perceived numerosity between adapting motion directions. The inter-trial interval was randomized between 1 and 1.5 s.

## Experiment 2: Motion direction-numerosity cross-adaptation with small numerosities

We also tested whether motion direction affects numerosity perception when small numerosities are used during adaptation. To this end, using the same basic paradigm as in Experiment 1, we adapted subjects with 30 dots and used a probe of 166 dots (instead of adapting with 400 dots and using a probe of 30 dots, as in Experiment 1). The test stimuli ranged from 53 to 357 dots (53, 63, 75, 89, 106, 126, 150, 178, 212, 252, 357). We also explicitly adapted subjects to random motion in both locations (upper and lower) and measured the resulting psychometric functions. However, as the two locations (test and probe) were equally adapted under these conditions, we did not observe an effect on the PSE in this condition (mean difference to probe=2.818 dots, $t(9)=0.486$, $p=0.639$). Subjects completed 220 trials per condition. The order of conditions was pseudorandomized across subjects and each condition was acquired on a separate day to prevent carry-over effects between motion directions. All other stimulus parameters were the same as in Experiment 1.

## Experiment 3: Direct effects of motion direction on numerosity perception in the absence of adaptation

In Experiments 1 and 2, we used an adaptation paradigm to reveal the effects of motion direction on numerosity perception. To test whether motion direction also affected numerosity perception in the absence of adaptation, we asked subjects to compare the numerosities of clouds of moving dots without showing adaptors. As in Experiments 1 and 2, we presented two clouds of moving dots (600 ms) in close temporal succession (ISI 400 ms) in the upper and lower half of the screen, respectively. One of the clouds contained dots moving at 100% coherence either to the left or to the right, with the number of dots ranging from 29 to 166 (29, 35, 42, 49, 59, 70, 83, 99, 118, 140, 166). The other cloud would always contain 70 dots moving coherently in random directions (excluding 80:100° and 260:280°). The direction of motion in this cloud would change every 100 ms to a new direction minimally differing 10° from the previous direction. We used dots moving coherently in random directions and not fully incoherent motion to isolate the effect of motion direction from possible effects of motion coherence, which were not of a concern in Experiments 1 and 2 where subjects judged the relative numerosity of two incoherently moving clouds of dots. On a randomly chosen half of the trials, the first, upper cloud contained coherent motion, and on the other half of the trials, the second, lower cloud contained coherent motion. Subjects had to indicate with a button press which of the two clouds appeared more numerous. Each motion direction was tested in a separate block of 220 trials.

## Experiment 4: Control for classical motion aftereffects

To test whether the differences in perceived numerosity after leftward and rightward motion arose as a direct consequence of adaptation, or whether the effect of motion direction on numerosity was mediated by the perception of a classical MAE, we conducted a control experiment in which we directly tested for the presence of classical MAEs in our paradigm. The design and all stimulus parameters of this experiment (*Figure 2—figure supplement 1*) were the same as for experiment 1, but subjects now had to report the perceived direction of motion instead of the numerosity of the dot clouds. Specifically, as in Experiment 1, we first adapted subjects in separate blocks to leftwards or rightwards motion, respectively, using 400 dots for 30 s above and below the fixation cross, respectively, followed by 5 s top-up adaptors (400 dots) on each trial. Subjects then saw two

consecutively presented clouds of dots, first at the upper and then at the lower location. The first, upper cloud (corresponding to the 'test' in Experiment 1) contained 60 incoherently moving dots, while the second, lower cloud (corresponding to the 'probe' in the main experiment) contained 30 dots that either moved to the left or to the right at 100% coherence. Subjects had to indicate by a button press whether they perceived the upper cloud to move in the same or a different direction than the lower cloud. If subjects consistently perceived a *repulsive* classical MAE, we expected them to respond 'same' after leftwards adaption when the lower cloud was moving to the right, and after rightwards adaptation when the lower cloud was moving to the left. If subjects consistently perceived an *attractive* classical MAE, we expected them to respond 'same' after leftward (rightward) adaptation when the lower cloud was also moving leftward (rightward). To assure that subjects were following task instructions, we also presented coherent motion to the left or right in the upper cloud on 1/3 of the trials, followed by motion in the same (50% of the trials) or opposite direction (50% of the trials) in the lower cloud. Here, we expected subjects to respond 'same' whenever the two clouds were actually moving in the same direction. In addition, to control for response biases, we also adapted subjects to incoherent dot clouds (0% motion coherence) above and below the fixation cross in a separate block. We used this condition to assess the subjects' overall propensity to response 'same', and later subtracted the individual percentage of 'same' responses from this condition from the percentage of 'same' responses on the conditions in which we expected attractive or repulsive motion for each subject. Subjects completed 60 trials in each block (random, right and left adaptor), and data from left and right adaptor were later pooled for analyses. The order of conditions was pseudorandomized, and, as in the main experiment, the random condition was tested between the two directional conditions to erase potential carry-over effects.

## Experiment 5: Contrasting coding schemes: two-pool opponent process vs. multichannel models

All stimulus parameters were the same as in Experiment 1 except the number of adapting dots in the 30 s adaptor and the 5 s top-up adaptor which was 30 instead of 400.

## Experiment 6: The reference frame of motion direction numerosity cross-adaptation

This experiment closely followed the design of Experiment 1, but we additionally manipulated the location of the dot clouds relative to the subject's gaze to probe in which reference frame motion direction-numerosity cross-adaption occurs (*Figure 4*). Subjects sat in a darkened experimental booth with their head positioned on chin rest and again initially adapted to a 30 s stimulus with directional motion (100% coherence) in the top cloud and random motion (0% coherence) in the bottom cloud at the beginning of each block, followed by a 5 s top-up adaptor on each trial (400 dots). Before we presented the test and probe stimuli, respectively, subjects had to make two consecutive saccades. The new saccade targets were indicated by moving the fixation cross to a new location on the screen (50% left, 50% right, randomized). Saccade targets were only updated once the eye tracker (Eyelink 1000, SR Research, Ottawa, ON, Canada; sampling rate 500 Hz) had registered that the subject was fixating at the indicated location, allowing us to equate the number of saccadic transients between conditions. After the second saccade had been completed, we presented the test stimulus (27, 38, 45, 54, 64 or 76 randomly moving dots), followed after 400 ms by the probe stimulus (30 randomly moving dots). In the baseline condition, subjects made a 14 dva rightward or leftward saccade, and then saccaded back to the initial fixation location at the center of the screen (*Figure 4a*). Hence, adaptor, test, and probe stimuli were presented at the same retinal and spatial location. This condition served to replicate the adaptation effect of motion direction on numerosity observed in Experiment 1 despite the intervening saccades. In the spatiotopic condition (*Figure 4b*), after adapting at the center of the screen, subjects first made a 14 dva rightward or leftward saccade, followed by a 28 dva saccade in the opposite direction. As in the baseline condition, test and probe stimuli were presented at their original location at the center of the screen, but with subjects fixating at the new location. Thus, adaptation and testing occurred at the same location in spatiotopic (or head-centered) coordinates, but not in retinotopic coordinates. In the retinotopic condition (*Figure 4c*), subjects also made a 14 dva leftward or rightward saccade, followed by a 28 dva saccade in the opposite direction, after adapting at the center of the screen. The test and probe stimuli

were then presented above and below the new fixation location; thus, they occurred at their original retinotopic coordinates, but at new spatiotopic coordinates. Finally, in the full field condition (*Figure 4d*), subjects made a 14 dva leftward or rightward saccade after adaptation, and then saccaded back to the original fixation location at the center of the screen. The test and probe stimuli were then presented in a location opposite of the final saccade direction at a new location on the screen that neither preserved retino- nor spatiotopic coordinates. Note that retinal eccentricity of the test/probe stimuli was the same in the 'same location' and the 'retinotopic' conditions (0 dva), as well as in the 'spatiotopic' and the 'full field' conditions (14 dva), respectively. Reference frames were probed on four separate days, and motion directions in consecutive sessions per day, each lasting for five blocks. The order of reference frames was pseudo-randomized across subjects, as was the order of motion directions. As in Experiment 1, an equivalent amount of random motion was presented between the directional conditions on every testing day to prevent carry-over effects.

## Analyses

Trials on which reaction times exceeded 2*median absolute deviation were excluded from the analyses. Psychometric functions were fitted in the Palamedes Toolbox (v1.7.0) (*Prins and Kingdom, 2009*) in Matlab (R2012b, The Mathworks, Natick, MA, USA), using a logistic function to obtain threshold and slope estimates per subject (fits using a cumulative Gaussian function yielded similar results). For Experiments 1, 2, and 3 threshold and slope estimates, respectively, were compared using paired *t*-tests (two-sided) in SPSS (v22, IBM, Armonk, NY, USA). Effect sizes were computed as Hedges' *d* (*Hedges and Olkin, 1985*), where values >0.8 indicate a large effect and values <0.5 indicate a small effect. For Experiment 4, we compared mean percentage 'same' responses using two-sided one-sample and paired *t*-tests; analyses of median percent 'same' responses using a percentile bootstrap procedure (*Wilcox, 2010*) yielded the same results. Given the sample sizes in Experiments 5 and 6, we resorted to non-parametric tests. Because we had a directional hypothesis (PSE right>left) in both experiments, planned comparisons were performed using one-sided exact Wilcoxon signed rank tests, but two-sided tests yielded identical patterns of results. In Experiment 6, threshold estimates were also entered into a non-parametric two factor repeated measures analysis of variance (rmANOVA) based on the aligned rank transform (*Higgins and Tashtoush, 1994*) using ARTool (v1.5.1) (*Wobbrock et al., 2011*) to assess interaction effects. Effect size was computed as partial $\eta^2$, where values >0.14 indicate a large effect and values <0.06 indicate a small effect. Spatial reference frames in Experiment 6 were determined by testing each experimental condition against 0. While it is standard in the literature to compare reference frames by contrasting conditions against 0 (*Arrighi et al., 2014*; *Knapen et al., 2009*; *Turi and Burr, 2012*), we note that such conclusions partially rest on non-significant results. We thus also performed Bayesian statistics, using an exact Wilcoxon signed rank test based on the prior ignorance Dirichlet process model (200000 samples, concentration parameter *s*=0.53) (*Benavoli et al., 2014*) to support our conclusions. All data are publicly available at Figshare (*Schwiedrzik et al., 2015*).

## Acknowledgements

This work was supported by a grant from the James S. McDonnell Foundation, awarded at the Latin American Summer School for Education, Cognitive, and Neural Sciences in San Pedro, Chile (to CMS), and a joint Summer Undergraduate Research Fellowship and Cognitive Science Research Award from Northwestern University (to BB). LM was supported by a Marie Curie International Outgoing Fellowship within the 7th European Community Framework Programme. We would like to thank all participants, in particular the faculty, of the LA Summer School for their invaluable input. Special thanks to David Poeppel, Clay Curtis, and Hakwan Lau for generously providing us with lab space and equipment, and to Daniel Spielsinger for help with data acquisition. We are indebted to Winrich Freiwald, Randy Gallistel, Alan Johnston, David Poeppel, the members of the Poeppel Lab, and the reviewers for thoughtful comments and discussions.

## Additional information

### Funding

| Funder | Grant reference number | Author |
|---|---|---|
| James S. McDonnell Foundation | Latin American Summer School for Education, Cognitive, and Neural Sciences | Caspar M Schwiedrzik |
| Northwestern University | Joint Summer Undergraduate Research Fellowship and Cognitive Science Research Award | Benjamin Bernstein |
| European Research Council | Marie Curie International Outgoing Fellowship within the 7th European Community Framework Programme | Lucia Melloni |

The funders had no role in study design, data collection and analysis, decision to publish, or preparation of the manuscript.

### Author contributions

CMS, Conceived the study, designed the experiment, analyzed the data, wrote the manuscript; BB, Acquired the data, designed the experiment, analyzed the data; LM, Acquired the data, designed the experiment, analyzed the data, wrote the manuscript.

### Author ORCIDs

Caspar M Schwiedrzik, http://orcid.org/0000-0003-0661-8859
Lucia Melloni, http://orcid.org/0000-0001-8743-5071

### Ethics

Human subjects: All subjects gave written informed consent before participation. All procedures were approved by The University Committee on Activities Involving Human Subjects at New York University.

## Additional files

### Major datasets

The following datasets were generated:

| Author(s) | Year | Dataset title | Dataset URL | Database, license, and accessibility information |
|---|---|---|---|---|
| Schwiedrzik CM, Bernstein B, Melloni L | 2015 | Motion along the mental number line: implications of shared representations for numerosity and space | http://dx.doi.org/10.6084/m9.figshare.1517750 | Publicly available on figshare |

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
