## [Decision Letter]

Thank you for submitting your work entitled "Motion along the mental number line" for peer review at *eLife*. Your submission has been evaluated by Timothy Behrens (Senior editor), a Reviewing editor, and two reviewers. We agree that there is great potential in this work. However, there are several issues that you will need to address before a final decision can be made regarding the suitability of this paper for *eLife*.

The reviewers have discussed the reviews with one another and the Reviewing editor has drafted this decision to help you prepare a revision and response.

The issues raised by one of the reviewers (and appended below) concern aspects of the procedures that make it difficult to interpret the results.

1) For 0% coherent motion, are dots replotted on each frame at random, or are they replotted in a random direction of motion with a lifespan of 166.6 ms? If it is the former, then all 0% stimuli differ in speed of apparent motion from the 100% stimuli.

2) Subjects are instructed to perform a two-interval comparison between the test (23-107) and probe (30) stimuli – thus only two test stimuli (23 and 27) are actually smaller than the probe. With a PSE of 52 dots (which is very close to the geometric mean that would be the predicted bisection point), how can the authors be confident that the subjects are actually comparing the test and probe, rather than ignoring the probe, and bisecting the test as "small" or "large"? To know if the probe is being processed, the authors should have used more than one probe to compare psychometric functions of comparisons to a smaller and larger probe (e.g. 30 and 60). The authors use a more typical test for the control experiment without an adaptor, with a probe of 70 and a test range of 29-166. However, the average PSEs for both probe-30 and probe-70 tasks can be explained by subjects performing the task, as if it were a one-interval bisection task.

3) Why did the MAE studies use 30 or 60 dots instead of the 400 used in the main study?

4) There should be a control condition in which random motion is presented in the location of the test stimulus. It would have been useful to see an unbiased PSE when the test location was adapted with 0% motion, and then compared that to the leftward and rightward for shifts. The psychometric functions reported in Figure 2 are both shifted to the right of an accurate PSE (30), which raises concerns about the effect of the adaptor on the processing of the test and probe stimuli, and shows both adaptors, leftward and rightward, causing an underestimation of test numerosity, with a small relative difference.

5) Figure 6 makes strong predictions about small numbers adaptors. If these results were obtained, they would suggest that the number of dots adapted leads to an over- or underestimation of test numerosity and that direction of motion modulates this main effect, but to a much smaller degree. The authors report in the supplemental materials that adapting with 30 dots instead of 400 fails to elicit an effect of motion direction on numerosity judgment (but do not report the PSEs of those experiments, which should be ~30 according to the model). To properly dissociate effects of adaptor number from adaptor direction on judgments, small and large numbers of dots should be used to adapt with both rightward and leftward motion (and random). In all conditions, a range of test stimuli should be used to make detection of leftward and rightward psychometric shifts possible to detect (e.g. 9-100 for a probe of 30).

---

## [Author Response]

*The issues raised by one of the reviewers (and appended below) concern aspects of the procedures that make it difficult to interpret the results. 1) For 0% coherent motion, are dots replotted on each frame at random, or are they replotted in a random direction of motion with a lifespan of 166.6 ms? If it is the former, then all 0% stimuli differ in speed of apparent motion from the 100% stimuli.*

The reviewer correctly points out that speed is an important variable, as it has been shown that it can directly affect numerosity perception (Afraz et al., 2004; Au and Watanabe, 2013). We thus made sure that the dot life time (166.6 ms) and the speed at which the dots traveled across the screen (4 dva/s) in our dynamic displays were the same in all conditions, including the random motion condition. The only difference between the directional and the random motion conditions was the level of motion coherence (100% vs. 0%, respectively). We have clarified this further in the Materials and methods section of our manuscript.

*2) Subjects are instructed to perform a two-interval comparison between the test (23-107) and probe (30) stimuli – thus only two test stimuli (23 and 27) are actually smaller than the probe. With a PSE of 52 dots (which is very close to the geometric mean that would be the predicted bisection point), how can the authors be confident that the subjects are actually comparing the test and probe, rather than ignoring the probe, and bisecting the test as "small" or "large"? To know if the probe is being processed, the authors should have used more than one probe to compare psychometric functions of comparisons to a smaller and larger probe (e.g. 30 and 60). The authors use a more typical test for the control experiment without an adaptor, with a probe of 70 and a test range of 29-166. However, the average PSEs for both probe-30 and probe-70 tasks can be explained by subjects performing the task, as if it were a one-interval bisection task.*

We use a two interval forced choice task in our experiments as it is a standard task in the field (e.g., Burr and Ross, 2008), but more importantly because it is thought to be virtually bias free (Egan, 1975; Green and Swets, 1974; Macmillan and Creelman, 2005; Wickens, 2002). However, when using the method of constant stimuli, as in our study, the range over which adaptation effects are evaluated has to be chosen in advance. Given the known, strong adaptation effects of numerosity independent of motion direction, it is customary to adjust the range of stimuli to the expected range of perceived numerosities (e.g., Liu et al., 2013), in order to maximize measurement sensitivity around the point of subjective equality. Our pilot experiments indeed confirmed the presence of static numerosity adaptation; and in light of the many conditions that we had to test, we opted to follow the procedures of other researchers in the field and to restrict the range of tested numerosities to 23-107 dots. The specific choice of adaptor and test/probe numerosities followed our pilot experiments and the original study on numerosity adaptation by Burr and Ross (2008).

Subjects can in principle solve the task either by taking the difference between the test and the probe intervals, or by comparing the test to an internal reference. However, both strategies are known to lead to similar points of subjective equality (PSEs; Morgan et al., 2000). It has been shown empirically that evidence for numerosity adaptation can be obtained with both procedures (Burr and Ross, 2008; Ross and Burr, 2010). If the subjects in our study had chosen to solve the task by comparing to an internal reference instead of by direct comparison of the test and the probe, this would mean that all our subjects explicitly disregarded their instructions, which seems highly unlikely. However, and this is the crucial point, neither strategy (direct comparison or comparison to a reference) predisposes or excludes that there is a difference between left and right motion directions, which is our main result. This is because the range of test stimuli is the same for all conditions. Hence, even if subjects disregarded their instructions and made a comparison to an internal reference, this cannot explain why there was a difference in PSEs between left and right motion direction.

In addition, the following control analysis provided further evidence against the reviewer’s suggestion that subjects were using an internal reference that could explain the differential effect of motion direction: We simulated ideal observers (n=11, same subject number as in the experiment 1) that would disregard the probe and instead use an internal reference to decide whether the test stimulus is bigger or smaller. The internal reference was computed as the running average over the last 10, 15, or 20 trials (the time scale over which such internal references are typically computed; Morgan et al., 2000) from the same trial sequences that our real observers saw in experiment 1. This yielded well-shaped psychometric functions, as expected. However, for none of the three running average window sizes we found a significant difference between left and right motion directions at *p*<0.05. Instead, PSEs for left and right motion directions were always virtually identical. Hence, using an internal reference to accomplish the task does not lead to the effects of motion direction we observed in experiment 1.

Finally, to address whether the PSE observed in the previous experiments could reflect the use of an internal reference constructed from the mean across test stimuli instead of an adaptation effect of static numerosity and/or texture density, we conducted a new control experiment in which we used an extended range of test stimuli, as suggested by the reviewer. In this new experiment, which we ran to test for the effect of small numerosity adaptors (30 instead of 400 dots, see response to the reviewer’s point #5 below), the probe stimulus contained 166 dots while the test stimuli included numerosities from 53 to 357 dots (53, 63, 75, 89, 106, 126, 150, 178, 212, 252, 357). The instructions were identical to experiment 1, i.e., to compare numerosities between test and probe stimuli. The average PSE was significantly different from the geometric mean of the test stimuli, which was 151 (mean difference 12.32 dots, *t*(9)=2.812, *p*=0.0203). This provides further evidence that subjects were compliant with the task instructions, and that the PSE does not simply reflect the use of an internal reference.

Thus, we find no evidence that subjects disregarded their instructions. Theoretically and empirically, solving the task by ignoring the instructions and using an internal reference does not lead to a difference between motion directions, which is the main result reported in the manuscript. We are thus confident that an alternative task strategy, i.e., use of an internal reference, cannot account for our results.

*3) Why did the MAE studies use 30 or 60 dots instead of the 400 used in the main study?*

This seems to be a misunderstanding. We apologize if the description of our control experiment was not sufficiently clear. The design and all stimulus parameters of the classical motion aftereffect control experiment were the same as for the main experiment, including the number of dots used to adapt (400 in both experiments). The only difference between the main experiments and the control experiment for classical motion aftereffects is that subjects had to report the perceived direction of motion instead of the numerosity of the dot clouds. After the adaptation phase, we presented clouds of 30 and 60 dots in order to closely mimic the main experiments during the test/probe phase. We have reformulated the description of this control experiment to clear up this point.

*4) There should be a control condition in which random motion is presented in the location of the test stimulus. It would have been useful to see an unbiased PSE when the test location was adapted with 0% motion, and then compared that to the leftward and rightward for shifts. The psychometric functions reported in*
Figure 2
*are both shifted to the right of an accurate PSE (30), which raises concerns about the effect of the adaptor on the processing of the test and probe stimuli, and shows both adaptors, leftward and rightward, causing an underestimation of test numerosity, with a small relative difference.*

The reviewer correctly points out that, in experiment 1, the PSEs for leftward and rightward motion are both shifted to the right, reflecting that in addition to a *relative difference* between left and right motion directions, which is the novel effect reported in this study, subjects generally underestimated numerosity. As discussed in the manuscript, this rightward shift of the PSEs is to be expected, given the known, strong, inevitable adaptation effects of numerosity (Burr and Ross, 2008) and/or texture density (Durgin, 1995, 2008) even in the absence of directional motion. This well-established effect is not a reason for concern, but an expected feature of numerosity and/or texture density adaptation (recently reviewed in Anobile et al., 2015). The novel finding here concerns the relative effect of leftward and rightward motion direction on numerosity perception, which is evident, e.g., in Figure 2 and Figure 5 (previously Figure 2 and Figure 4). As the directional motion conditions were matched for numerosity and texture density, the adaptation effects of static numerosity and/or texture density cannot account for the difference in motion directions that we observed and only the difference in motion direction can explain motion direction-numerosity cross-adaptation in our paradigm.

Regarding an additional control condition in which subjects are adapted to random motion, we note that this condition is already included into our paradigm *by design*. Specifically, subjects compare two locations, one that is adapted to directional motion, and one that is equally adapted to random motion. Hence, the PSE already reflects a comparison to the control condition that the reviewer requests (left vs. random and right vs. random). However, following the reviewer’s request, we did include a condition with random motion in a new control experiment in which we used smaller adaptor numerosities (see response to the reviewer’s point #5 below for details). As the two locations (test and probe) are equally adapted under these conditions, we did not observe an effect on the PSE in this condition (mean difference to probe = 2.818 dots, *t*(9)=0.486, *p*=0.639).

Regarding the difference between left and right motion directions, we note that the effect is statistically significant (*p*=0.029), of considerable statistical effect size *(d*Hedges=0.953, where values >0.8 indicate a large effect and values <0.5 indicate a small effect), reproducible over several experiments (now 4 different studies bringing the total number of tested subjects to 55), and observed for large and small adaptors. It is evident in Figure 2 that the effect can range up to about 20 dots in individual subjects (also see Figure 2—figure supplement 3, previously Figure 2), exhibiting a very large effect size. Most importantly, an effect of motion direction on numerosity perception is present even in the absence of adaptation, when a rightward shift in the PSEs is of no concern (Figure 2, control experiment for the effect of motion direction in the absence of adaptation). For all these reasons, we are confident that the effect of motion direction on numerosity perception exists, although the average difference between left and right motion direction (7.12 dots in experiment 1) may appear small at first sight.

*5) Figure 6 makes strong predictions about small numbers adaptors. If these results were obtained, they would suggest that the number of dots adapted leads to an over- or underestimation of test numerosity and that direction of motion modulates this main effect, but to a much smaller degree. The authors report in the supplemental materials that adapting with 30 dots instead of 400 fails to elicit an effect of motion direction on numerosity judgment (but do not report the PSEs of those experiments, which should be ~30 according to the model). To properly dissociate effects of adaptor number from adaptor direction on judgments, small and large numbers of dots should be used to adapt with both rightward and leftward motion (and random). In all conditions, a range of test stimuli should be used to make detection of leftward and rightward psychometric shifts possible to detect (e.g. 9-100 for a probe of 30).*

We completely agree with the reviewer that our results reflect two main effects. The first, main effect the reviewer mentions can be seen in Figure 2: subjects generally underestimate numerosity after adaptation with 400 dots. This effect likely reflects previously reported adaptation effects of numerosity (Burr and Ross, 2008) and/or texture density (Durgin, 2008). In addition, there is a second main effect, also visible in Figure 2: motion direction has a *differential effect* on perceived numerosity, whereby rightward motion leads to stronger aftereffects than leftward motion. This is in line with the model depicted in Figure 6: the rightward shift of the colored lines reflects the first main effect the reviewer mentions (due to static numerosity and/or texture density), while the distance between leftward (orange) and rightward (green) motion reflects the differential effect of motion direction. The main novel result of our investigations is the presence of this second main effect, the effect of motion direction, which cannot be accounted for the effects of static numerosity and/or texture density adaptation.

We interpret our data to result from the differential activity in two pools of neurons; one that increases its response with increasing numerosity and rightward motion, and another one that increases its activity with decreasing numerosity and leftward motion. The differential activity of these two pools as a consequence of adaptation leads to perceptual aftereffects. The reviewer correctly points out that the model depicted in Figure 6 can further be extrapolated from large to small adapters. Specifically, one would predict that when small numbers are used for adaptation, the small number/leftward motion pool should be more strongly activated (and thus more adapted) by leftward motion than by rightward motion. Hence, one would now expect the PSE after leftward motion adaptation to be smaller than after rightward motion adaptation (see new Figure 6).

To test these predictions we conducted a new experiment (n=11, 4 female, mean age 27.1 yrs, range 22-35 yrs, 9 right handed; one subject had to be excluded from data analyses due to an apparent failure to do the task) where we adapted with 30 dots and used a probe of 166 dots (instead of adapting with 400 dots and probing with 30 dots). We did not use even smaller numbers, as we wanted to avoid counting or subitizing of small numbers as reasonable strategies to solve the task. As requested by the reviewer, we extended the range of test stimuli to include numerosities ranging from 53 to 357 dots (53, 63, 75, 89, 106, 126, 150, 178, 212, 252, 357). In addition, also following the reviewer, we adapted subjects to random motion in both locations and measured the resulting psychometric functions. The order of conditions was pseudorandomized across subjects and each condition was acquired on a separate day to prevent carry-over effects between motion directions. All other stimulus parameters were the same as in experiment 1.

As predicted by our model we observed that the PSE after leftward motion adaptation with 30 dots is smaller than after rightward motion adaptation with 30 dots (Figure 7 below and Figure 2 in the revised manuscript, all data points falling above the unity line; mean difference in PSE 15.85 dots; *t*(9)=4.523, *p*=0.001, *d_Hedges_*=1.017). Thus, motion direction has differential effects on numerosity perception both when adapting with small (30, new experiment) and when adapting with large quantities (400, original experiments). As in experiment 1, there were no effects on slopes (mean difference -0.123, *t*(9)=-0.755, *p*=0.469, *d_Hedges_*=-0.313).

Author response image 1.Motion direction-numerosity adaptation when adapting with small numerosities.Leftward motion cross-adaptation leads to a stronger overestimation effect than rightward motion adaptation. The scatter plot shows that PSEs for leftward motion were consistently smaller than for rightward motion (mean difference in PSE 15.85 dots; *t*(9)=4.523, *p*=0.001, *d_Hedges_*=1.017, two-sided). This indicates that as when adapting with large numerosities, leftward motion shifts numerosity perception down the number line, while rightwards motion shifts numerosity perception up the number line. A general main effect of static numerosity and/or texture density leading to overestimation independent of motion direction is also evident in a majority of subjects (all data points falling left and/or below the dashed lines).**DOI:**
http://dx.doi.org/10.7554/eLife.10806.013

In addition, Figure 7 shows that in a majority of subjects, PSEs were shifted to the left of the probe (166 dots, all data points falling left and/or below the dashed lines), reflecting overall overestimation, the second expected main effect. This effect of (static) numerosity and/or texture density adaptation has previously been reported in Burr and Ross (2008); see their Figure 2. The results of the new control experiment the reviewer suggested thus confirm the presence of two main effects: a static effect of numerosity and/or texture density and a differential effect of motion direction, in accordance with the model we proposed in Figure 6.

Together, the experiments with large adaptors and small tests/probes originally reported in the manuscript and the new experiment with small adaptors and large tests/probes show that the two main effects are well dissociated: when adapting with large numbers, the static adaptation effect shifts the PSE to the right (underestimation). In contrast, when adapting with small numbers, static adaptation shifts the PSE to the left (overestimation). Under both scenarios, there is an additional effect of motion direction: rightward motion increases the perceived numerosity of dots. Leftward motion has the opposite effect: decreasing the perceived numerosity of dots. These differential effects of motion direction are in accordance with an upward or downward shift on the mental number line, respectively.

We now report the results of our new control experiment in the main text (including a new Figure 2). We have also expanded Figure 6 to include the reviewer’s predictions about small number adaptors. We would like to thank the reviewer for suggesting this control experiment, as it allowed us to generalize our findings to small numerosities, further underscoring the robustness of motion direction-numerosity cross-adaptation. We hope that the controls analyses and control experiments, as well as further clarifications of our stimulation protocol addressed the outstanding points to the reviewers’ and editors’ satisfaction.